# Effectiveness Evaluation Method of Application of Mobile Communication System Based on Factor Analysis

**DOI:** 10.3390/s21165414

**Published:** 2021-08-10

**Authors:** Guohui Jia, Jie Zhou

**Affiliations:** Information and Communication Institute, National University of Defense Technology, Changsha 410073, China; zhoujie8912@163.com

**Keywords:** factor analysis, mobile communication system, effectiveness evaluation method

## Abstract

The application mode of army mobile communication networks is closely related to combat mission and application environment. Different combat missions and application environments result in different network structures and different service priorities, which requires a semi-automatic system to support the network scheme design. Therefore, evaluating the efficiency of network schemes generated by automatic planning is a problem that needs to be urgently addressed. In the past, researchers have proposed a variety of methods to evaluate the effectiveness of mobile communication systems, most of which are based on simulation methods and ignore the historical data of network usage. This paper studies an effectiveness evaluation method of mobile communication network design schemes and proposes a design scheme for the evaluation and optimization of network plans. Furthermore, the improved method of effectiveness evaluation based on factor analysis is discussed in detail. The method not only effectively uses historical data but also reduces the amount of data collection and calculation. In order to adapt to the preference requirements of different task scenarios, a decision preference setting method based on cluster analysis is proposed, which can render the output optimization result more reasonable and feasible.

## 1. Introduction

Mobile communication systems can ensure command and control and other information transmission in army mobile warfare [1,2,3]. It can connect various reconnaissance systems, information processing systems, command and control systems and weapon systems scattered across land, sea, air and outer space in order to realize situation sharing and collaborative control among various elements of the battlefield [4,5,6]. Different mobile communication systems support different combat missions and serve different objects, and their network structures; interconnection modes and monitoring modes are also different [7,8,9,10]. The mobile communication system organization of the army synthetic division or brigade troops in complex terrain environments generally utilizes ultra-shortwave radio, shortwave radio, trunked mobile communication system and satellite communication. They can form a multi-layer network in order to meet the needs of information transmission of different levels of users. Figure 1 shows a network structure of a real war-game performed in Northern China of a synthetic brigade. This Figure is based on the static network loop model of Xue et al. [11] and Jeffrey R. Cares [12].

At the same time, the network structure will be changed according to the combat mission, the number of participating units, terrain restrictions, the number of equipment and other conditions. Different network structures will also result in different network planning parameters, such as network routing, access mode, subnet frequency distribution, encryption mechanism and key distribution method. For example, in Figure 1, the broadcast protocol is selected in detachment network 1, while detachment network 2 selecs peer-to-peer protocol to form a chain network, and detachment network n selects the Carrier Sense Multiple Access (CSMA) protocol in order to form a fully connected network. Therefore, designing and planning the network completely based on human efforts will be very complicated.

We study a semi-automatic network planning assistant optimization system for the mobile communication network of army divisions or brigade troops. As shown in Figure 2, the system is connected with network planning equipment and network management equipment of mobile communication system. The system uses multi-agent system technology to simulate and evaluate the network design schemes generated by network planning equipment and finally generates the preferred ranking scheme for network planning equipment [13]. The modules related to the content of this paper include three agents (Observer, Resolver and Analyzer) and a historical database.

The observer is used to receive and preprocess data and is the interface of external information input. Its main task is to receive the network design schemes and preference setting input by network planning equipment and to export the schemes to the Analyzer. In addition, it also receives the real operation data of the actual operation process of the network imported from the management equipment and exports the operation data to the Resolver after simple preprocessing.

The Resolver is used to analyze historical data, and it is also the implementation part of the effectiveness evaluation method based on factor analysis in this paper. It mainly carries out factor analysis on historical data in order to determine the key parameters and weights.

The Analyzer is used to simulate and deduce the network design schemes. It mainly deduces the schemes and obtains a little bit of statistics according to the key parameters determined by the Resolver. After each scheme is simulated, quantitative calculation is carried out according to the weight determined by Resolver and the schemes are sorted according to the preferences.

Based on the above network planning assistant optimization system design, the mobile communication system networking scheme needs to be evaluated quickly and recommended in the order of the decision makers’ preferences. Therefore, there is an urgent need for a fast, objective and favorite-supporting network effectiveness evaluation method [14,15,16,17].

The general definition of effectiveness is that the system or equipment can achieve the expected use target ability under the specific use conditions and within the specified time [18,19]. However, the mobile communication system faces many tasks, which presents some characteristics such as diversity, complexity and science. Many scholars have performed relevant research and proposed evaluation methods and ideas for communication systems under different tasks and application scenarios. Some effective and typical evaluation methods and models can be summarized as follows: generalized index method; principal component analysis method [20,21]; analytic hierarchy process (AHP) method [22,23]; fuzzy comprehensive evaluation method [24,25,26]; grey correlation analysis evaluation method [27]; neural network [28,29,30]; and ADC method [31]. These methods have certain effects on effectiveness evaluation and are usually used with different synthetic strategies to solve different types of problems. For example, the reference [32] evaluated the security of the LTE-R communication system with AHP and the fuzzy comprehensive evaluation method and achieved the security status objectively and reasonably. Aiming at the multicriteria decision making (MCDM) problems, reference [33] used the fuzzy grey Choquet integral (FGCI) method to evaluate MCDM problems with many interactive and qualitative indexes; this method can reduce the influence of experts’ subjective preference. The reference [34] combines AHP, grey correlation technique and TOPSIS to solve the problem of green decoration materials selection, this method can obtain the weight vector of hierarchical index structure reasonably. However, the evaluation index system is difficult to construct in complicated problems. The reference [35] uses the data envelopment analysis (DEA) model to describe undesirable indicators and to obtain more reasonable results with Shannon’s entropy in order to assess the operational efficiency of public bus transportation. Furthermore, when considering the communication system effectiveness evaluation, the large number of influencing factors and their complex relationship renders this type of work difficult. The reference [36] used an analytic network process (ANP) to aggregate and calculate complex indicators in order to solve the problem that the dependency of elements within the hierarchy and the feedback influence of lower elements on upper elements should be considered in a complex system. The reference [37] proposed a robust mobility management scheme for (TCN) and developed a physical test-bed environment. It used service availability, network throughput and first recover packet arrival time in order to evaluate the effectiveness of the management scheme. The reference [38] proposed a fast algorithm for performance analysis for optimizing tactical communications networks in a timely fashion. For each class of traffic, this paper discussed three measures and analytical algorithms for performance ability evaluation, such as average call-arrival rate, average call-holding time and service priority. The reference [39] proposed a unified T&E framework for the test and evaluation of the tactical communication networks. It studied a visual and modular radio software programming tool that supports easy and fast test and evaluation ranges from simulations to the use of over-the-air wireless testbeds and hardware-in-the-loop network emulation tests.

Therefore, it can be observed that there is currently no unified and fixed mode for the evaluation of tactical mobile communication networks, and the selection of evaluation methods and indicators is largely related to the tasks undertaken; however, simple and rapid evaluation methods are focuses of research. In addition, most of these studies used the method of simulation, statistically calculated the indexes according to the comparison matrix or weight [40,41,42] and then evaluated the advantages and disadvantages of the system according to the comprehensive score results [43,44]. None of these approaches takes full advantage of the historical data of network operation. These historical data, which are recorded through previous exercises and task implementation, possess real network guarantee situations and task completion effect evaluation, and the historical possess great reference value. However, the amount of historical data is very large, and the evaluation conclusion must be generated quickly in the process of network scheme design and optimization; thus, we have to convert the large amount of data into fewer and easier to calculate indicators and algorithms. Factor analysis can transform a large amount of data into a small number of independent factors. These independent factors can reflect most of the information of the original data, which is within the scope of a reasonable research idea.

Factor analysis is a multivariate statistical method that extracts public factors from the evaluation data set and describes the original variables with public factors [45]. It is an objective evaluation method completely starting from the data set, which can better overcome the influence of personnel’s subjective consciousness and improve the objectivity and science of the evaluation conclusion [46].

The factor analysis model is defined as a linear model showing the relationship between common factors and variables. In general, the factor analysis model is shown with Equation (1).
(1)$$\begin{array}{c} x_{1}=a_{11}F_{1}+a_{12}F_{2}+\cdots +a_{1m}F_{m}+\varepsilon_{1}\\ x_{2}=a_{21}F_{1}+a_{22}F_{2}+\cdots +a_{2m}F_{m}+\varepsilon_{2}\\ \cdots \\ x_{p}=a_{p1}F_{1}+a_{p2}F_{2}+\cdots +a_{pm}F_{m}+\varepsilon_{p} \end{array}$$

The matrix form is expressed as follows:(2)X=AF+ε
where $x_{1}$, $x_{2}$… $x_{p}$ represents random data of *p* original indexes and $F_{1}$, $F_{2}$… $F_{m}$ are the extracted common factor scores that are unobservable random variables. *ε* is the special factor and represents the part that cannot be explained by common factors. The coefficient *a_ij_* is the factor loading and its size indicates the importance of the common factor relative to the original indexes [47].

Data dimensionality reduction cannot reduce the collection of original data but also brings in the complexity of calculation. However, the mobile communication system networking scheme needs to be evaluated quickly and recommended in the order of the decision makers’ preferences. Therefore, there is an urgent need for a fast, objective and favorite-supporting network effectiveness evaluation method.

The main contributions of this paper can be summarized as follows:We study an effectiveness evaluation method of mobile communication network design schemes and propose a design scheme for the evaluation and optimization of network plans.We propose an improved method of effectiveness evaluation based on factor analysis. It can not only effectively use the historical data but also greatly reduce the amount of data collection and calculation.We propose a decision preference setting method based on cluster analysis.

The rest of the paper is structured as follows: Section 2 presents the background for our paper. Section 3 introduces the traditional effectiveness evaluation method of factor analysis. Section 4 proposes an improved method of effectiveness evaluation based on factor analysis and verifies its validity by an example. Section 4 proposes a decision preference setting method based on cluster analysis. Section 5 Synthesizes the above studies and provides the detailed process of effectiveness evaluation of mobile communication network design schemes. Section 6 concludes this study and mentions future research implications.

## 2. Evaluation Indices and Data Acquisition

### 2.1. Evaluation Indexes

This paper mainly takes a certain brigade mobile communication system as the research object and selects eight representative indexes in order to evaluate the system’s operation efficiency. The specific evaluation indexes and definitions are shown in Table 1.

### 2.2. Data Acquisition

In the kinds of army synthetic division/brigade tactical communication networks shown in Figure 1, the network management equipment can be accessed at any node in the network the individual desires. The architecture is similar to the MIML-IDPS System [48], which can form a 3 tier or 2 tier management structure. The 3 tier network management structure includes a first-level network management center, several second-level network management centers and third-level node devices. The first-level network management center is responsible for controlling all nodes in the network and can divide the network into different administrative areas, which are managed by the second-level network management center. The last level devices are the backbone network nodes and the detachment network access nodes. These nodes act as network management agents, which can control the instructions of network management devices to specific devices and also report the status of devices to network management devices.

We collected 31 groups of real data from participating units in several war-games performed in Northern China. The amount of data obtained is very large, and it is necessary to preprocess the data before evaluating network performance.

### 2.3. Data Preprocessing

According to the definition of evaluation indexes in Table 1, the calculation model of each index is as follows.

(1) The information encryption intensity reflects the degree of correlation between the encrypted information and the original information. In this study, the weighted sum of the three parameters of channel encryption algorithm complexity, encryption key length and key update cycle was calculated. The calculation model is shown as follows:(3)$$X1=\omega_{1}\frac{\alpha_{cpl}}{3}+\omega_{2}\frac{\alpha_{len}}{4}+\omega_{3}\frac{t_{\max}-t_{cyc}}{t_{\max}-t_{\min}}\;\;\;\omega_{1}+\omega_{2}+\omega_{3}=1$$
where *α_cpl_* represents the category of encryption algorithm, three different algorithms (DES, 3DES and AES) can be selected and the corresponding assigned value is from 1 to 3. *α_cpl_* represents the length of the encryption key, which can be selected as 56, 64, 128 and 256, and the corresponding values are assigned 1–4. *t_cyc_* represents the key update cycle and *t*_min_ and *t*_max_ represent the longest and shortest cycle of key update, which are 2 and 24 in this paper. ω_1_, ω_2_, ω_3_ represent the weight of the three indexes, which can be artificially specified or obtained by AHP and other methods [49].

(2) The network throughput reflects the data transmission volume of the whole network per unit time. In this study, in order to remove the influence caused by the different number of users, the total amount of information sent by all nodes is counted and then divided by the total number of users. The calculation model is shown as follows:(4)$$X2=\frac{1}{nT}\sum_{i=1}^{n}C_{output^{i}}$$
where $C_{output^{i}}$ represents the information sent by node *i*, *n* is the total number of nodes in the network and *T* is the total working time.

(3) The end to end average delay reflects the efficiency of information transmission from each source node to the sink node. Since there are different levels of users in a tactical mobile network, the influence of information delay loss among different levels of users is different. The influence of information transmission delay between backbone node networks in network operation is related to its importance in the network. Data processing in this paper is divided into two types of nodes. For the Command and Control (C2) network and detachment network nodes, the weighted calculation is carried out according to the level of source and sink nodes. For the backbone nodes, the weighted calculation is carried out according to the link numbers of source and sink nodes. The calculation model is shown as follows.

(5)$$X3=\omega_{1}\sum_{i=1}^{n_{C2}+n_{De}}\sum_{j=1}^{ren_{i}}\frac{s_{s}^{j}+s_{d}^{j}}{16}t_{{}_{delay}}^{j}+\omega_{2}\sum_{i=1}^{n_{Bk}}\sum_{j=1}^{ren_{i}}\frac{c_{s}^{j}+c_{d}^{j}}{2\times c_{\max}}t_{{}_{delay}}^{j}\;\;\;\omega_{1}+\omega_{2}=1$$$n_{C2}$ represents the number of C2 nodes, $n_{De}$ represents the number of access nodes in detachment network, $n_{BK}$ represents the number of backbone network nodes, $ren_{i}$ represents the total number of data packets received by node *i*, $s_{s}^{j}$ represents the level of the source node of the packet *j* and $s_{d}^{j}$ represents the level of the sink node of the packet *j*. According to the level of troop establishment, there are eight levels altogether; thus, the maximum level of transceiver node sum is 16. $t_{delay}^{j}$ represents the delay of the packet *j*, $c_{s}^{j}$ represents the connection number of the packet *j* source node, $c_{d}^{j}$ represents the connection number of the packet j sink node and $c_{\max}$ represents the maximum connection number of the backbone node. *ω*_1_ and *ω*_2_ represent the weight of two types of networks, which can be artificially specified or obtained by AHP and other methods.

(4) The network state perception reflects the ability of network management equipment to perceive changes when network node states change, and it is weighted by the sum of accuracy and delay. Accuracy is expressed as the number of times the network management device has successfully identified the network state change divided by the number of times that the network agent senses the change and triggers the trap reports. Delay is represented as the time lag between the node device sending trap notification and the network management device identifying the change. The calculation model is shown as follows.
(6)$$X4=\omega_{1}\frac{n_{success}}{n_{trap}}+\omega_{2}\frac{1}{n_{success}}\sum_{i=1}^{n_{success}}\frac{t_{sp\max}-t_{spdelay}^{i}{}_{}}{t_{sp\max}-t_{sp\min}}\;\;\;\omega_{1}+\omega_{2}=1$$
$n_{success}$ represents the number of times the network management device has successfully identified the changes, $n_{trap}$ represents the number of times the node device has sent trap notifications, $t_{spdelay}^{i}$ represents transmission delay of the state change notification *i* and $t_{sp\max}$, $t_{sp\min}$ represent the maximum and minimum values of the state change notification transmission delay. *ω*_1_ and *ω*_2_ represent the weight of accuracy and delay, which can be artificially specified or obtained by AHP and other methods.

(5) The timeliness of network adjustment reflects the delay between a network adjustment instruction being sent and the successful network adjustment. Contrary to the delay calculation method in the previous article, this index calculates the time lag between the network management device sending the adjustment instruction and the node network agent processing and outputting the control instruction. The calculation model is shown as follows.
(7)$$X5=\frac{1}{n_{nm}}\sum_{i=1}^{n_{nm}}\frac{t_{nm\max}-t_{nmdelay}^{i}{}_{}}{t_{nm\max}-t_{nm\min}}$$
$n_{nm}$ represents the number of network adjustment instructions, $t_{nmdelay}^{i}$ represents the transmission delay of the network adjustment instruction *i* and $t_{nm\max}$, $t_{nm\min}$ represents the maximum and minimum values of transmission delay.

(6) The success rate of temporary network access reflects the success rate of temporary network entry. The calculation method was the detachment nodes’ ratio of successful network access to total application times. Successful network access means that the detachment node receives the network access permission message sent by the network management device. The calculation model is shown as follows.
(8)$$X6=\frac{n_{access}}{n_{apply}}$$
$n_{access}$ represents the number of detachment nodes that has been successfully accessed. $n_{apply}$ represents the number of applications.

(7) The voice interruption rate reflects the effect of voice communication. Similarly to the end-to-end average delay, the influence of voice interruption loss among different levels of users is different. The level of nodes on both sides of the call should also be considered in the calculation. The calculation model is shown as follows.
(9)$$X7=\frac{1}{n_{C2}+n_{De}}\sum_{i=1}^{n_{C2}+n_{De}}\left(\frac{1}{{\mathrm{ncall}}_{i}}\sum_{j=1}^{{\mathrm{ncall}}_{i}}\frac{s^{i}+s_{d}^{j}}{16}CF(j)\right)$$
$n_{C2}$ represents the number of C2 nodes, $n_{De}$ represents the number of access nodes in detachment network, $ncall_{i}$ represents the total number of calls from node *i*, $s^{i}$ represents the level of node *i* and $s_{d}^{j}$ represents the user level of the *j*th call. According to the level of troop establishments, there are eight levels. altogether Thus, the maximum level of transceiver node sum is 16. *CF(j)* is an index function, which is 1 when the call *j* failed or interrupted and 0 for the rest.

(8) The voice link building time reflects the average time from dialing to successful chain establishment, and it is calculated by the weighted sum of the two levels of the caller. The calculation model is shown as follows.
(10)$$X8=\frac{1}{n_{C2}+n_{De}}\sum_{i=1}^{n_{C2}+n_{De}}\left(\frac{1}{{\mathrm{ncall}}_{i}}\sum_{j=1}^{{\mathrm{ncall}}_{i}}\frac{s^{i}+s_{d}^{j}}{16}t_{calldelay}^{j}\right)$$
$n_{C2}$ represents the number of C2 nodes, $n_{De}$ represents the number of access nodes in detachment network, $ncall_{i}$ represents the total number of calls from node *i*, $s^{i}$ represents the level of node *i* and $s_{d}^{j}$ represents the user level of the *j*th call. According to the level of troop establishments, there are eight levels altogether. Thus, the maximum level of transceiver node sum is 16. $t_{calldelay}^{j}$ represents the time of chain establishment for the call *j*.

## 3. An Improved Effectiveness Evaluation Method Based on Factor Analysis

### 3.1. Traditional Factor Analysis

By using this process, we have obtained the evaluation data for each application of the Tactical Mobile Communications System. We collected 31 groups of real data from participating units in several war-games performed in Northern China. In analyzing the eight evaluation indexes, data standardization is required prior to the evaluation because the dimensions and numerical ranges expressed are different.

Data standardization includes assimilation processing and dimensionless processing [50,51]. The assimilation processing of indexes is also called forward processing [52]. In the evaluation, the larger the value of some indexes, the better the effectiveness is. In some situations, the larger the value of some indexes, the worse the effectiveness is [53]. Moreover, the closer some indexes are to the middle range, the better the effectiveness is and the more they are above or below the middle range, the worse the effectiveness is [54].

We first process the collected evaluation data into the assimilation matrix. Then, the Z-core method is adopted to conduct dimensionless processing on the data set, and the data set shown in Table 2 is obtained. All calculation and analysis diagrams in this paper are produced by R of version 3.6.3.

The KMO value is 0.788 ≥ 0.5 and Bartlett value is 0.000 ≤ 0.05, which meets the criteria, indicating that the data set is suitable for factor analysis.

From the data set in Table 2, the eigenvalues of the correlation coefficient matrix are 5.1925, 1.2606, 0.6471, 0.3883, 0.2346, 0.1462, 0.0879 and 0.0428. By calculating the variance contribution rate of the factor, we observe that the cumulative variance contribution rate of the first three factors is more than 85%. Therefore, the first three components were selected as the main factors for factor analysis.

According to the eigenvectors corresponding to the selected factors, the non-rotated factor loading matrix on the left side of Table 3 is generated.

Factor loading is one of the important indexes in factor analysis, and it is the data source of factor naming. It can be observed from the Table 3 that the loading of factor 1 on most indexes is large, which indicates that factor 1 is an important comprehensive factor. However, some indexes, such as *X*7, have larger loads on factor 1 and factor 2, which cannot directly explain the factors. It is necessary to rotate the factor load matrix so that indexes can only show a larger load on one factor and a smaller load on other factors.

The orthogonal rotation method of maximum variance is used to rotate the data to obtain the factor loading on the right side of Table 3.

It can be observed from the factor load matrix after rotation in Table 3 that factor 1 has larger loads on indexes *X*2, *X*3, *X*7 and *X*8, which indicates that factor 1 mainly reflects the characteristics of four indexes: network throughput, end to end average delay, voice interruption rate and voice link building time. According to the specific meaning and characteristics of these indexes, factor 1 is named as the information transmission factor.

Factor 2 has a larger load on indexes *X*4, *X*5 and *X*6, which indicates that factor 2 mainly reflects the characteristics of three indexes: ‘network state perception’, ‘timeliness of network adjustment‘ and ‘success rate of temporary network access‘. According to the specific meaning and characteristics of these indexes, factor 2 is named as network control factor.

Factor 3 has a larger load on index *X*1, which indicates that factor 3 mainly reflects the characteristic of information encryption intensity. According to the specific meaning of the index, factor 3 is named as security protection factor.

Next, we used the Bartlett factor scoring method to calculate the score of the rotated factor and obtained Table 4.

The factor score is the sum of the product of the standardized index value and the score coefficient. For example, the score of factor 1 is as follows:(11)F1=0.044136×X1+0.243241×X2+0.360151×X3−0.347005×X4−0.219985×X5−0.021879×X6+0.382698×X7+0.437015×X8

Other factor scores were calculated similarly and scores of each factor can be obtained. By using the following equation to calculate the overall score, the calculated results are shown in Table 2.
(12)$$Score=\sum_{i=1}^{m}\frac{\varphi_{i}}{\phi }\cdot S_{i}$$
$S_{i}$ is the factor score, $\varphi_{i}$ is the factor variance contribution rate and ϕ is the cumulative variance contribution rate of factors.

### 3.2. Analysis on the Importance of Evaluation Index

The square sum of the loads of the *i*th index on all m factors is called the commonness of the index [55], which reflects the effect of the index on all factors, that is, the importance of each original index. The calculation expression is shown as follows.
(13)$$H_{{}^{i}}=\sum_{j=1}^{m}a_{ij}^{2}\;\;\;i=1,2,\cdots ,p$$
*a_ij_* is the load of the *i*th index on the *j*th factor, *m* is the number of factors and *p* is the number of original indexes.

By comparing the commonness of the indexes, we can observe which variable plays a greater role. Therefore, the commonality of all variables can be normalized to the weight of the variable.
(14)$$W_{i}=\frac{H_{i}}{\sum_{j=1}^{p}H_{j}}$$
$H_{i}$ is commonness of the index *i* and *p* is the number of original indexes.

According to the factor loading matrix of Table 3, the weight of each index can be calculated as shown in Table 5.

According to the data in Table 5, the top four indexes in terms of weight are as follows: Voice link building time, information encryption intensity, end to end average delay and network state perception. By observing the factor load after rotation in Table 5, it can be observed that the load of the ‘voice link building time’ index is 0.9032, 0.3425 and 0.1302, respectively. This means that this index has a higher influence on the three factors. Similarly, the reason why *X*3 and *X*4 have higher weights is that they have higher influence factors among the three factors. The load of *X*1 index in three factors is 0.0833, 0.1625 and 0.9299, which means that this index has a higher influence on the second and third factors.

### 3.3. Correlation Analysis

Previously, factor 1 has been named as the information transmission factor, which is mainly determined by index *X*2, *X*3, *X*7 and *X*8. Factor 2 is named as the network control factor, which is mainly determined by index *X*4, *X*5 and *X*6. Factor 3 is named as the security protection factor, which is mainly determined by index *X*1.

In order to ensure that there is no information overlap between the selected indexes, the correlation analysis of the main indexes of each factor is further screened. The correlation coefficients of factors 1 and 2 are shown in Table 6 and Table 7.

The correlation between *X*3 and other indexes is greater than the critical value (0.5), which indicates that the end-to-end transmission delay (*X*3) of the system has a strong correlation with the network throughput (*X*2), voice interruption rate (*X*7) and voice link building time (*X*8). Moreover, the correlation coefficient between the end-to-end transmission delay (*X*3) and voice link building time (*X*8) is the largest. In fact, these two indexes are the measurements of system transmission from two different scenarios. The end-to-end transmission delay is the average transmission delay of fixed length packets between any node in the network. The voice of the system studied is based on VoIP, and the time from call initiation to successful chain construction is roughly equal to the sum of the transmission delay of call instruction packets between source and destination. Therefore, it has a strong correlation. Considering that the voice link establishment time (*X*8) includes the end-to-end data transmission process from the link establishment process, additionally, the comprehensive weight of index *X8* in the importance analysis of the last section is the largest, and the influence on the three factors are large. Thus, the index *X*8 is retained and the index *X*3 is discarded.

Similarly, the correlation between *X*2 and *X*8 is also large, and the correlation coefficient is 0.877. In fact, *X*2 reflects the total data transmission volume per unit time of the network. The larger the total data transmission volume of the network, the busier the lines between the nodes of the network, and the longer the packet queue in the nodes. As a result, the transmission and forwarding time of each packet increases when the voice call link is established; thus, the voice link establishment time (*X*8) also increases. In addition, in the importance analysis of the previous section, the comprehensive weight of index *X*8 is the largest; thus, the index *X*8 is retained and the index *X*2 is discarded. After analysis, the simplified indexes of information transmission factor indexes are determined as *X*7 and *X*8.

The correlation coefficients between *X*4, *X*5 and *X*6 are greater than the critical value (0.5), indicating that there is a great correlation between the network situation awareness, the timeliness of network state adjustment and the success rate of temporary network access. In fact, the network state awareness measures the correctness and data transmission delay of the network management agent in each network device when the network state changes. The timeliness of network adjustment is that the network management equipment sends the network adjustment instruction data to the network management agent, and the network management agent outputs the adjustment instruction to the device controller in order to implement the device control. It can be considered as the reverse information transmission and control of network perception information, and thus it has strong relevance. The success rate of temporary network access measures the success rate of new network users sending network access applications to the network management device, which assigns the corresponding parameters to them, and the device network management agent receives and outputs them to the device controller to implement the device control and to finally complete the network access. Therefore, the actual data transmission and control process is similar to the network state perception. In addition, considering that the comprehensive weight of the index *X*4 in the importance analysis of the previous section is larger, therefore, the index *X*4 is retained and the indexes *X*5 and *X*6 are discarded. After analysis, the simplified index of network control factor is *X*4.

### 3.4. Simplified Algorithm Model

By the analysis of the importance and correlation of the indexes, the evaluation indexes of operational effectiveness of mobile communication system can be simplified as *X*1, *X*3, *X*7 and *X*8, and the comprehensive weight of these four indexes needs to be adjusted to *W_i_*^*^.
(15)$$W_{i}^{*}=\frac{W_{i}}{\sum_{j=1}^{k}W_{j}}\;\;\;i=1,2,\cdots ,k$$
*W_i_*^*^ is the new normalized commonness of the index *i*, *W_i_* is the original normalized commonness of the index *i* and *k* is the number of simplified indexes.

According to the weight of each index and the standardized data set, the weighted sum can obtain the total score of each unit.
(16)$$Score_{i}^{*}=\sum_{j=1}^{k}W_{i}^{*}.Z_{ij}\;\;\;i=1,2,\cdots ,n$$
$Score_{i}^{*}$ is the simplified index score of each unit, $Z_{ij}$ is the normalized data of each unit and *n* is the number of units.

By calculating the score values of each unit and arranging them by the simplified algorithm model, the comparison with the ranking results of factor comprehensive scores in Table 2 is shown in Table 8.

Table 8 shows that the maximum deviation between the evaluation ranking of the simplified algorithm model and the ranking results obtained by the previous factor analysis method is 3, in which 42% (13 units) of the evaluation ranking possessed no change, 39% (12 units) of the evaluation ranking possessed one difference, 13% (4 units) of the evaluation ranking possessed two differences and 6% (2 units) of the evaluation ranking possessed three differences. This shows that the evaluation conclusion of the simplified algorithm model is reliable and ideal. In the first stage, the method can be used to evaluate historical data and to calculate simplified algorithmic models. In the second stage, the simplified indexes are used to evaluate the simulation data of the design scheme, which greatly reduces the workload of data recording, statistics and calculation, greatly facilitates the implementation of the late evaluation and has strong economic benefits.

This method can be applied to the semi-automatic network planning assistant optimization system of mobile communication network of army division/brigade troops. The latest simplified indexes and weights can be obtained by using the historical data and stored in database.

When network planning needs to be performed again, the network plan set generated by the planning equipment is imported into the system, and the system simulates all the network plans. It only requires a small amount of simulation data to be collected according to the simplified index and can directly calculate the effectiveness of each network plan according to the simplified model.

After each network operation, data will be imported into the system through the network management device so as to update the indexes and weights and to keep the system updated iteratively.

## 4. Preference Strategy

By using the factor scores and ranking in Table 8, we can produce specific evaluations on the operational efficiency of a mobile communication system of each participating unit. The result is a comprehensive ranking result, which is suitable for most cases, and the performance of all aspects of the network scheme is relatively balanced. In the actual application process, more emphasis may be placed on the efficiency of information transmission in some cases. In other cases, more emphasis may be placed on the perception or security of the network. Therefore, it is very important to propose preference strategies under different requirements. We first analyze the data results of this paper and then propose a preference strategy method.

### 4.1. Analysis of Evaluation Results

In this paper, we selected the most commonly used system clustering method [56], which is to divide n samples into several categories. The system clustering method first clusters the samples or variables as a group, then it determines the statistics of similarity between class and class, and then it selects the closest two or several classes merged into a new class of computing similarity between a new class and other kinds of statistics. Finally, it then chooses the one closest to the group of two or several groups merged into a new class until all of the samples or variables are merged into a class. K-means clustering algorithm is an iteratively solved clustering analysis algorithm. It requires the specification of the number of data groups K, it then calculate the distance between each object and each seed clustering center, and then it assigns each object to the nearest clustering center.

Due to the variety of data sources, it is difficult to determine the appropriate group number in advance, and the selection of appropriate classification value results in the inaccuracy of clustering results and affects the accuracy of the subsequent recommendation algorithm. In the selection of the system clustering method, results can be presented by clustering, and the user can select the appropriate number of groups, which is convenient to improve the accuracy of the subsequent recommendation algorithm.

(1)First, n samples are regarded as one category;(2)The distance between categories is calculated;(3)Select the two categories with the smallest distance to merge into a new category;(4)Repeat steps 2 and 3 to reduce one category at a time until all samples become one category.

A Euclidean distance is used to calculate the distance between two samples.
(17)$$d_{{}^{i,j}}=\sqrt{\sum_{t=1}^{m}{\left(x_{it}-x_{jt}\right)}^{2}}$$
$d_{i,j}$ is the distance between sample *i* and *j*, $x_{it}$ and $x_{jt}$ are the scores of samples *i* and *j* on the factor *t*.

The shortest distance method is used to calculate the distance *D_pq_* between category *G_p_* and *G_q_*.
(18)$$D_{pq}=\min(d_{i,j})\;\;\;i\in G_{p},j\in G_{q}$$

According to the information transmission factor (factor 1), we can see that units 19, 9, 26 and 13 have higher scores in factor 1. From Figure 3, the distance between these four units and other units in factor 1 is also large, which indicates that the communication systems of these three units possess large data transmission capacity, small delay, low voice interruption rate, strong battlefield communication ability and good voice communication performance. The rest of the units are roughly divided into four levels of the following: 1, 2, 11, 18, 20 and 25 are the second levels; 10, 12, 14, 21, 22 and 23 are the third levels; 5, 6, 7, 15, 16, 24, 29 and 31 are the fourth levels; 3, 4, 8, 27, 28 and 30 are the fifth levels.

According to the network management and control factor (factor 2), we can observe that units 1, 11, 2 and 9 have higher scores in factor 2. From Figure 4, it can be observed that the distance between the four units and other units in factor 2 is also large, which indicates that the four units have strong battlefield control abilities when using mobile communication systems, can better grasp the whole network operation state and can timely adjust the network as needed. They can support members’ random access and temporary access requests. The remaining units are roughly divided into four levels of the following: 3, 10, 15, 22, 27, 28 and 30 are the second levels; 4, 18, 23 and 29 are the third levels; 5, 6, 7, 8, 12, 13, 16, 17, 19, 20, 21, 24, 25 and 31 are the fourth levels; 14 and 26 are the fifth levels.

According to the security protection factor (factor 3), we can observed that unit 26 and 1 have higher scores in factor 3. From Figure 5, the distance between these two units and other units in factor 3 is also large, which indicates that these two units have strong security protection ability, high network robustness, strong information encryption and can be used to transmit information with higher security levels. The remaining units are roughly divided into four levels of the following: 5, 6, 7, 9, 15, 25 and 28–31 are the second levels; 4, 8, 10, 17, 22 and 24 are the third levels; 3, 12, 13, 18 and 23 are the fourth levels; and 2, 14, 16, 19–21 and 27 are the fifth levels.

From the ranking of comprehensive scores, it can be observed that the comprehensive scores of 1, 9, 19 and 11 are the highest. From Figure 6, it can be observed that these four units are also far away from other units; that is to say, the comprehensive application effect of the mobile communication systems of these four units is the best. The remaining units are roughly divided into four levels of the following: 2, 10, 13, 15, 22, 25 and 26 are the second levels; 20, 23 and 29–31 are the third levels; 3–8, 12, 17, 24, 27 and 28 are the fourth levels; and 14, 16 and 21 are the fifth levels.

### 4.2. Preference Selection Algorithm

According to the results of the above cluster analysis, we can choose four preference strategies: information transmission, network control, security protection and comprehensive scores. According to the comprehensive scores, the best performances are observed in units 1, 9, 19 and 11. The comprehensive score of unit 19 ranks third, with the highest score of factor 1, but the scores of factor 2 and factor 3 are very low, which means that the information transmission ability of unit 19 is very strong while the network control ability and security protection ability are poor. Therefore, the optimization of network design scheme should be based on a certain preference and refer to other factors.

The preference selection algorithm proposed in this paper is shown as follows.

(1)According to the preference strategy (information transmission, network control, security protection and integration), perform cluster analysis on the evaluation results and obtain n-level data set (the number of grades can be specified by users based on the cluster analysis results);(2)Select the first two levels of the corresponding cluster analysis as initial ranking;(3)Delete schemes at the lower 2 levels among other preferences;(4)Output the remaining schemes in order as the result.

Based on the above algorithm, when the preference of comprehensive scores is selected, the optimal results are 1, 9, 11, 10 and 22. When the preference of information transmission is selected, the optimal results are 9, 1 and 11. When preference of network control is selected, the optimal results are 1, 11, 9, 10 and 22. When preference of security protection is selected, the optimal results are 1, 9 and 15. Compared with the traditional factor analysis comprehensive score method, the preference selection algorithm proposed in this paper can avoid other low scores except when considering the factors of user preference or comprehensive scores, rendering the output optimization result more reasonable and feasible.

## 5. A Summary of the Evaluation Process

Based on the above analysis, the effectiveness evaluation overall process of mobile communication design schemes includes 13 steps as shown in Figure 7. The steps 1–9 correspond to the improved algorithm described in Section 3. The steps 11–13 correspond to the preference strategy algorithm described in Section 4. By examining the above case analysis results, the method proposed in this paper with hybrid solutions and algorithms has better performance when applied on mobile communication system. Firstly, the indicators reduce by 50% which the amount of data collection and calculation decreases. Secondly, from Section 4, we can obtain different results of preference selection, the preference strategy algorithm can make specific decisions for different scenarios and the method can ensure that the calculation speed meets the requirements of dynamic mobile communication systems on the battlefield.

## 6. Conclusions

This paper proposed a solution for the rapid effectiveness evaluation of multiple schemes, which can be adopted to army division/brigade troops’ mobile communication network planning equipment. The solution includes a semi-automatic network planning assistant optimization system, an improved simplification factor evaluation method and a preference strategy algorithm based on cluster analysis. By investigating an example that includes eight indicators, the improved simplified factor evaluation method can reduce the indicators by 50% on the premise of maintaining the accuracy of evaluation. The preference strategy algorithm can firstly ensure the preference and delete other low correlation items to increase the feasibility of the optimization results. The analysis results show that although the solutions and algorithms are based on the research of the army division/brigade mobile communication network, the solution can also be applied to the field of network planning evaluation of mobile communication systems in various industries that require to be deployed on demand and be quickly established. For future works, we plan to further expand the scope of indexes (only eight indexes are selected in this paper and 152 indexes are commonly used in the actual system) and to verify the effectiveness, adaptability and stability of this method. We will develop a prototype system based on the methods of this paper and we plan to develop multi-agent simulation technology. In the future, we hope to further combine the artificial intelligence method to apply network scheme evaluation to the network scheme generation stage in advance.

## Figures and Tables

**Figure 1 sensors-21-05414-f001:**
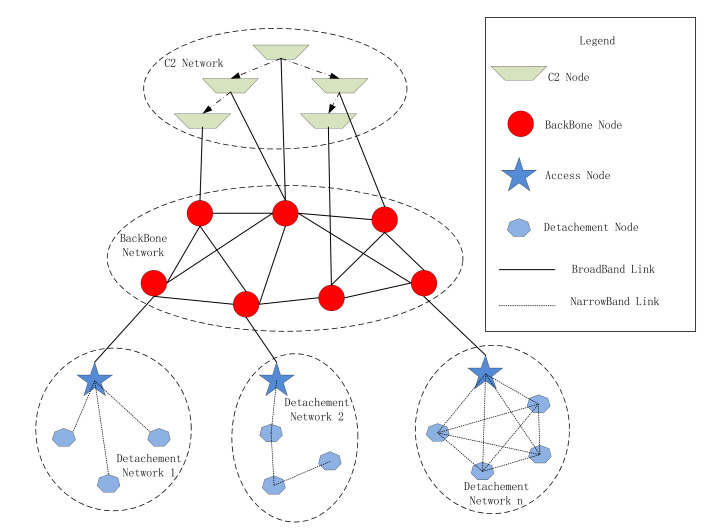
Structure of mobile communication network of a brigade.

**Figure 2 sensors-21-05414-f002:**
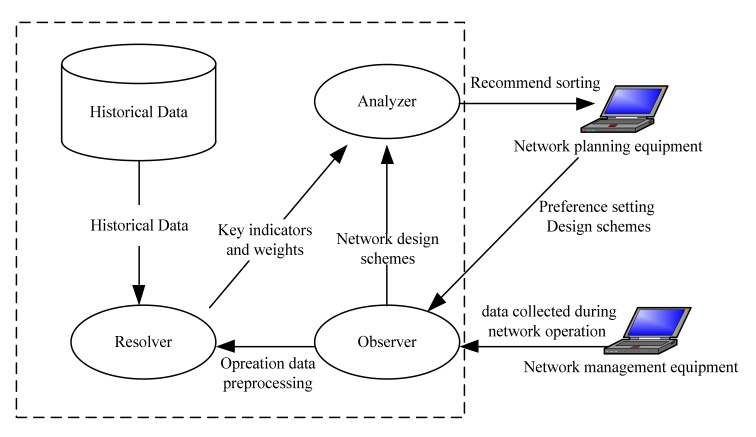
Network planning assistant optimization system.

**Figure 3 sensors-21-05414-f003:**
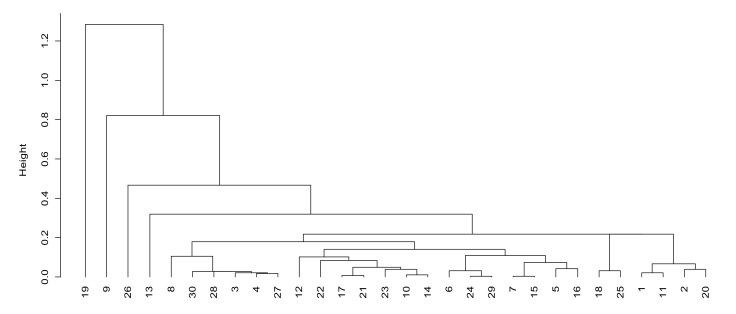
Cluster analysis of information transmission factors.

**Figure 4 sensors-21-05414-f004:**
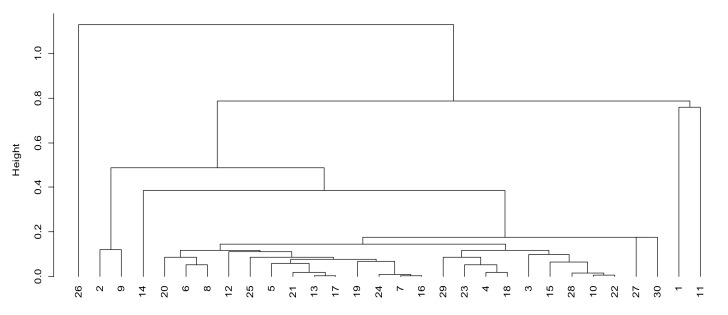
Cluster analysis of network management and control factors.

**Figure 5 sensors-21-05414-f005:**
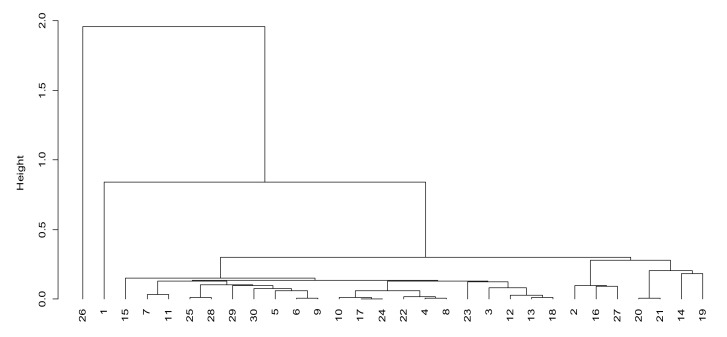
Cluster analysis of security protection factors.

**Figure 6 sensors-21-05414-f006:**
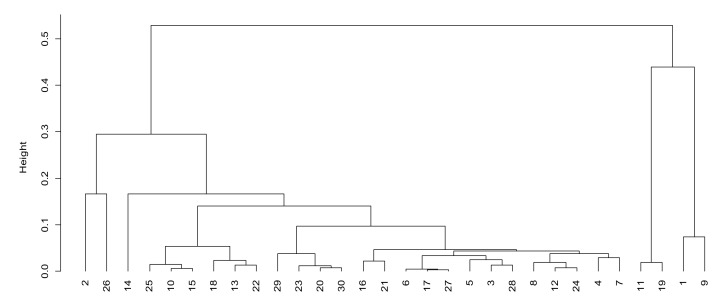
Cluster analysis of comprehensive score.

**Figure 7 sensors-21-05414-f007:**
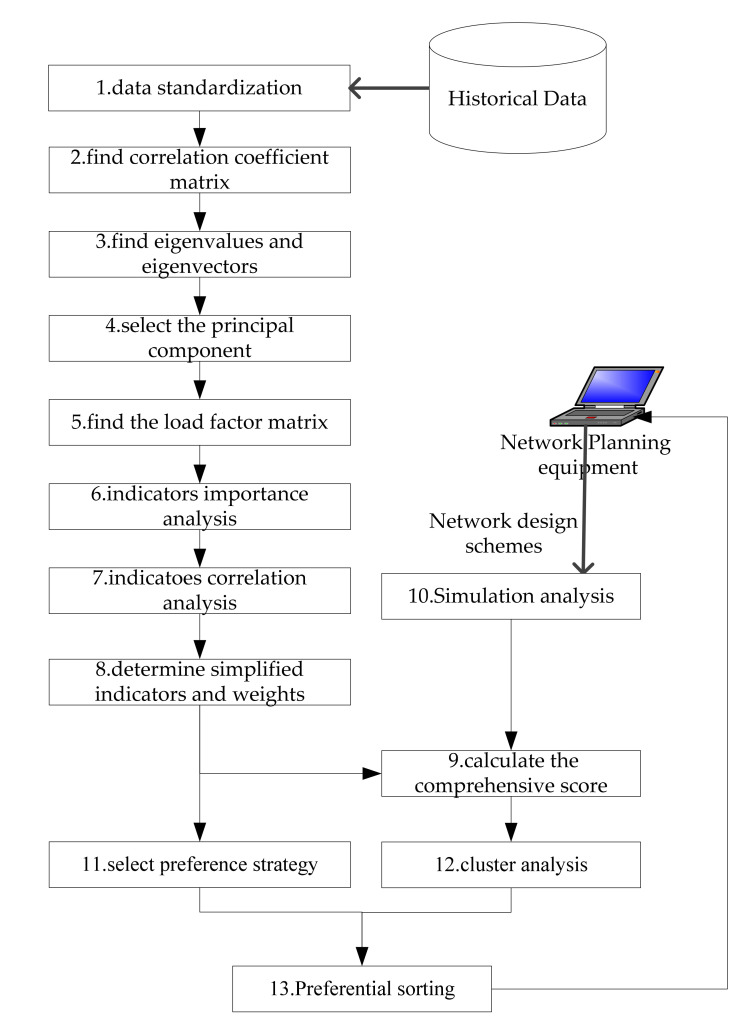
The overall process of effectiveness evaluation.

**Table 1 sensors-21-05414-t001:** Table of Evaluation Indexes.

Index Name	Index Definition	Abbreviation
Information encryption intensity	Average value of correlation coefficient between enciphered information and original information.	*X*1
Network throughput	The data transmission volume of the whole network per unit time.	*X*2
End to end average delay	The average value of the delay from the source node to the destination node.	*X*3
Network state perception	When the state of the network node changes, the accuracy and delay weighted value of the network management device perception changes.	*X*4
Timeliness of network adjustment	The time delay from the network adjustment instruction to the successful network adjustment	*X*5
Success rate of temporary network access	Success rate of temporary users.	*X*6
Voice interruption rate	The ratio of the number of voice communication interruptions to the total number of calls.	*X*7
Voice link building time	The average time of voice users from dialing to successful chain building.	*X*8

**Table 2 sensors-21-05414-t002:** Parameter value, factor score and comprehensive score of each participating unit after standardization.

Unit	*X*1	*X*2	*X*3	*X*4	*X*5	*X*6	*X*7	*X*8	Factor 1	Factor 2	Factor 3	Total Points
1	1.960	2.493	1.909	3.130	2.297	3.084	0.361	1.699	0.602	2.938	1.639	1.644
2	−0.098	0.347	0.861	0.871	2.063	0.860	1.613	0.689	0.534	1.392	−0.947	0.582
3	−0.112	−0.950	−0.779	0.729	−0.257	−0.807	−0.360	−0.394	−1.006	0.429	−0.216	−0.339
4	−0.197	−0.445	−1.058	0.051	−0.725	−0.516	−0.759	−0.795	−0.983	0.024	0.006	−0.437
5	0.348	−0.222	−1.041	−0.682	−0.864	−0.598	−0.687	−0.395	−0.409	−0.756	0.551	−0.364
6	0.328	−0.635	−0.351	0.338	−0.977	−0.485	−0.640	−0.631	−0.679	−0.174	0.485	−0.285
7	0.009	−0.654	−0.669	−0.155	−1.062	−0.680	−0.402	−0.645	−0.533	−0.520	0.138	−0.408
8	−0.098	−0.910	−0.816	0.107	−1.051	−0.656	−0.526	−0.792	−0.859	−0.227	0.001	−0.474
9	0.225	2.570	3.203	2.014	1.127	2.794	1.182	2.292	2.235	1.270	0.490	1.570
10	−0.141	0.008	0.218	−0.416	0.907	0.001	−0.491	0.124	−0.115	0.262	−0.075	0.030
11	0.876	1.508	1.352	1.778	2.748	1.563	1.137	1.260	0.622	2.179	0.169	1.111
12	−0.563	−0.985	−0.101	−1.277	−0.665	−0.442	−0.668	−0.709	−0.267	−0.868	−0.297	−0.493
13	−0.276	−0.102	0.964	−0.548	0.252	−0.386	0.656	0.544	0.942	−0.678	−0.337	0.120
14	−1.366	−1.066	−0.773	−1.791	−0.866	−0.848	0.015	−0.833	−0.126	−1.254	−1.226	−0.737
15	1.098	−0.259	−0.424	−0.135	0.445	0.362	−0.473	−0.278	−0.528	0.331	0.796	0.024
16	−0.438	−1.009	−1.039	−0.404	−0.632	−1.101	0.715	−0.894	−0.453	−0.517	−0.852	−0.548
17	0.262	−0.815	−0.427	−0.934	−0.549	0.032	0.351	−0.693	−0.065	−0.674	−0.088	−0.292
18	0.040	0.006	−0.192	−0.125	0.091	0.567	0.784	0.073	0.277	0.042	−0.324	0.084
19	−1.156	2.052	1.680	0.470	0.638	1.131	3.425	3.399	3.521	−0.596	−1.410	1.130
20	−1.297	−0.379	−0.152	−0.828	0.208	0.056	1.226	−0.023	0.495	−0.313	−1.621	−0.180
21	−2.394	−0.215	−0.063	−0.915	−0.921	−0.691	−0.367	−0.620	−0.057	−0.696	−1.614	−0.570
22	0.309	−0.485	0.452	−0.074	0.374	0.664	0.223	−0.105	0.027	0.267	−0.015	0.107
23	−0.398	−0.387	0.035	−0.389	0.093	−0.069	0.032	−0.443	−0.164	−0.029	−0.460	−0.168
24	−0.423	−0.930	−0.510	−0.864	−0.469	−0.704	−1.085	−0.497	−0.643	−0.529	−0.085	−0.501
25	−0.074	0.616	0.073	0.258	−0.783	−0.398	−0.089	0.034	0.245	−0.429	0.309	0.010
26	3.238	0.921	0.925	−1.137	−1.059	−0.820	−0.884	0.924	1.411	−2.383	3.598	0.415
27	−0.725	−0.431	−0.770	0.179	0.487	−0.204	−0.409	−0.527	−0.965	0.782	−0.761	−0.289
28	−0.056	−0.188	−0.689	0.181	−0.438	−0.412	−1.147	−0.778	−1.033	0.247	0.296	−0.326
29	−0.094	0.959	−0.442	0.302	−0.532	−0.406	−1.303	−0.551	−0.648	0.130	0.646	−0.131
30	0.187	−0.183	−0.814	0.634	0.142	−0.622	−1.146	−0.131	−1.063	0.606	0.411	−0.188
31	1.026	−0.231	−0.562	−0.366	−0.022	−0.270	−0.284	−0.303	−0.317	−0.258	0.793	−0.096

**Table 3 sensors-21-05414-t003:** Factor loading matrix.

Index	Nonrotation			After Rotation		
	Factor 1	Factor 2	Factor 3	Factor 1	Factor 2	Factor 3
*X*1	0.320650	0.843998	0.287874	0.083278	0.162547	**0.929873**
*X*2	0.909567	0.120463	0.172870	**0.725623**	0.488718	0.326088
*X*3	0.911390	−0.071196	0.242330	**0.837270**	0.401471	0.179513
*X*4	0.780331	0.293705	−0.430681	0.228377	**0.881250**	0.227832
*X*5	0.827407	−0.010898	−0.395764	0.396441	**0.826709**	−0.027115
*X*6	0.933679	0.030354	−0.169476	0.586084	**0.737937**	0.115563
*X*7	0.656920	−0.646705	0.113004	**0.791305**	0.218188	−0.434478
*X*8	0.913760	−0.152106	0.303287	**0.903241**	0.342474	0.130169

**Table 4 sensors-21-05414-t004:** Factor score coefficient table.

Index	Factor 1	Factor 2	Factor 3
X1	0.044136	−0.155257	0.789911
X2	0.243241	−0.063296	0.219128
X3	0.360151	−0.171547	0.122934
X4	−0.347005	0.631644	−0.022096
X5	−0.219985	0.550707	−0.218813
X6	−0.021879	0.313877	−0.050131
X7	0.382698	−0.146096	−0.376688
X8	0.437015	−0.252909	0.101338

**Table 5 sensors-21-05414-t005:** Weight table of each index.

	*X*1	*X*2	*X*3	*X*4	*X*5	*X*6	*X*7	*X*8
*H_i_*	0.8980	0.8717	0.8944	0.8807	0.8413	0.9014	0.8625	0.9501
*W_i_*	**0.1265**	0.1228	0.1260	**0.1240**	0.1185	0.1270	**0.1215**	**0.1338**
*W_i_* ^*^	**0.2501**	--	--	**0.2452**	--	--	**0.2402**	**0.2501**

**Table 6 sensors-21-05414-t006:** The correlation coefficient of information transmission factor indexes.

	*X*2	*X*3	*X*7	*X*8
***X*2**	1.0000	0.8494	0.4655	0.8770
***X*3**	0.8494	1.0000	0.6019	0.8866
***X*7**	0.4655	0.6019	1.0000	0.7204
***X*8**	0.8770	0.8866	0.7204	1.0000

**Table 7 sensors-21-05414-t007:** Correlation coefficient between management and control factor indexes.

	*X*4	*X*5	*X*6
***X*4**	1.0000	0.7023	0.7688
***X*5**	0.7023	1.0000	0.8149
***X*6**	0.7688	0.8149	1.0000

**Table 8 sensors-21-05414-t008:** Comparative analysis of simplified index and original method.

Unit	Factor Total Score	Factor Ranking	Simplified Index Score	Simplify Sorting	Sort Difference
1	1.644	1	2.1178	1	0
2	0.582	5	0.8886	5	0
3	−0.339	22	−0.3680	20	−2
4	−0.437	25	−0.5566	25	0
5	−0.364	23	−0.5136	23	0
6	−0.285	18	−0.3784	21	3
7	−0.408	24	−0.5300	24	0
8	−0.474	26	−0.5914	26	0
9	1.57	2	1.9360	2	0
10	0.03	10	0.0236	11	1
11	1.111	4	1.5177	3	−1
12	−0.493	27	−0.6742	28	1
13	0.12	7	0.1394	9	2
14	−0.737	31	−0.9431	31	0
15	0.024	11	0.0410	10	−1
16	−0.548	29	−0.6078	27	−2
17	−0.292	20	−0.3476	19	−1
18	0.084	9	0.1539	8	−1
19	1.13	3	1.4659	4	1
20	−0.18	16	−0.1548	14	−2
21	−0.57	30	−0.7751	30	0
22	0.107	8	0.1690	7	−1
23	−0.168	15	−0.1948	15	0
24	−0.501	28	−0.6824	29	1
25	0.01	12	−0.0421	12	0
26	0.415	6	0.2849	6	0
27	−0.289	19	−0.3079	18	−1
28	−0.326	21	−0.4422	22	1
29	−0.131	14	−0.2590	17	3
30	−0.188	17	−0.2417	16	−1
31	−0.096	13	−0.1267	13	0

## Data Availability

Not applicable.

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
