# Peer review of "Effectiveness Evaluation Method of Application of Mobile Communication System Based on Factor Analysis"

_sensors, 2021, doi:10.3390/s21165414_

Round 1

Reviewer 1 Report

Sensors-1279739-Peer-Review-V1

Paper Summary:

The paper studies an effectiveness evaluation method of mobile communication network design schemes and puts forward a design scheme to evaluate and optimize the network plans. It also discusses the details of the improved method of effectiveness evaluation based on factor analysis. Furthermore, the study proposes a decision preference setting method based on cluster analysis to adapt to the preference requirements of different task scenarios. Overall, the authors have tried to present an improved preference selection algorithm, discuss the summary of the evaluation process, and conclude their study. In this regard, the reviewer found the work vital and sound. Generally, the manuscript includes most of the primary sections required for scientific research papers and well-structured that the contents are written in a clear way that is easy to understand the core points. However, to ensure the quality of the article, the following suggestions and comments are needed to be considered and addressed to improve the research further.

Therefore, the reviewer found some issues that need improvement and forwarded them to the authors as major and minor issues.

Major issues:

  1. The reviewer recommends the authors follow the structure of the sections as per the MDPI Sensor manuscript format. Hence, authors are suggested to combine the two sections, namely “Introduction” and “Background,” together without losing the flow of ideas of the paper as “Introduction.” 
  2. Authors are recommended to have a separate table that shows the list of notations and its description to make all the symbols used in the paper easy to understand and clear for the reader.
  3. In some equations, the authors haven’t defined the variables where they are used. For instance, please check the definitions of variables in equations 12 and 13 onpage 10, 14 & 15 on page 12 and 16 on page 13. In addition, some acronyms are used before they are defined in the paper. Please try to correct and update them accordingly.
  4. There are also typos, formatting, and grammatical errors in several sections of the paper. Hence, authors should recheck, paraphrase the contents, and proofread the entire manuscript.
  5. In this paper, the authors prefer the system clustering method to divide nsamples into several categories. Why you prefer this method to other clustering methods? Briefly include why you choose it in section 5 on page 14 and state the comparative advantages of the method used in your study. Besides, please describe the effectiveness of the improved algorithm (preference selection algorithm) you proposed compared with the traditional benchmark algorithms used in this paper.
  6. According to the standard formatting layout guide of MDPIfor authors, it is suggested that the conclusion section shouldn’t have many paragraphs. Therefore, please try to compress the conclusion section of your study into one paragraph (which is common).

Minor issues:

  1. The reviewer suggests the authors check the title of the paper to make it sound and straightforward. For instance, it will be more sound if you paraphrase it as: “Evaluation Methods of Application Effectiveness of Mobile Communication System Based on Factor Analysis.”, or Effectiveness Evaluation Method of Application of Mobile Communication System based on Factor Analysis.” or other title which is clear and not vague for the reader.
  2. In the part of the main contributions of this paper on page 2, the authors have presented their contributions separately. However, there is the repetition of words/phrases. So, the authors should avoid the repetition of words or phrases by enumerating the contributions using list items briefly as follows: 
  • We study an effectiveness evaluation method mobile of communication network design schemes and put forward a design scheme to evaluate and optimize network plans.
  • We propose an improved method of effectiveness evaluation based on factor analysis.
  • We present a decision preference setting method based on cluster analysis.
  1. Some of the figures used in the paper need improvements to make the figures visible and more descriptive for the reader, including inserting border lines, space between the text body, and another standard figure formatting. Example: Refer to Figure 3, update the figure on page 3, and change the caption of Figure 7 on page 16as: “The overall process of effectiveness evaluation.”
  2. Generally, the reviewer recommends the authors strictly follow all the MDPI manuscript preparation guidelines specific to theSensor 

Reviewer 2 Report

This work presents an improved effectiveness evaluation method of mobile communication system based on factor analysis. It seems interesting, but the following comments are suggested to improve its presentation.

  1. Some grammar and sentences needs to be improved, e.g., in line 329, “The calculation expression is as follows” should be corrected as “The calculation expression is shown as follows”.
  2. The literature review seems is organized poorly. The authors mainly list the factor analysis methods, e.g., AHP and grey correlation analysis. But the topic of this work is mobile communication system evaluation, thus related new and important papers need to be updated, the following works may be considered, e.g., Fuzzy Grey Choquet Integral for Evaluation of Multicriteria Decision Making Problems With Interactive and Qualitative Indices,” IEEE Transactions on Systems, Man, and Cybernetics: Systems, doi: 10.1109/TSMC.2019.2906635; “Target disassembly sequencing and scheme evaluation for CNC machine tools using improved multiobjective ant colony algorithm and fuzzy integral,” IEEE Transactions on Systems Man Cybernetics - Systems, 2019, 49(12): 2438-2451; “Green decoration materials selection under interior environment characteristics: a grey-correlation based hybrid MCDM method,” Renewable & Sustainable Energy Reviews, 2018, 81(1): 682-692; "Physical Safety and Cyber Security Analysis of Multi-Agent Systems: A Survey of Recent Advances," in IEEE/CAA Journal of Automatica Sinica, vol. 8, no. 2, pp. 319-333, February 2021; "Performance Evaluation of Public Bus Transportation by Using DEA Models and Shannon's Entropy: An Example From a Company in a Large City of China," in IEEE/CAA Journal of Automatica Sinica, vol. 8, no. 4, pp. 779-795, April 2021; and "Safety Evaluation of High Speed Railway LTE-R Communication System Based on AHP and Fuzzy Comprehensive Evaluation," 2019 International Conference on Intelligent Transportation, Big Data & Smart City (ICITBS), 2019, pp. 211-214.
  3. What is the superiority of the proposed model? Moreover, please discuss the complexity of the model in terms of the number of variables.
  4. The author needs more reasons to explain why adopt such 8 factors (X1-X8).
  5. When you process your variable, e.g. X1 (based on Equation 3) is obtained by AHP, whose is a subjective method. This phenomenon is against with author statement, see abstract, and please give more explanation in detail.
  6. With respect to the motivation to present a new model, a clearer justification in needed taking into account the limitations of the previous work.
  7. The comparisons of the new model are not well justified. More discussion of the results is needed
  8. In the conclusion section, I can find out that these sentences are mentioned in the previous sections. The main scientific value of the research should be mentioned in this section.

Reviewer 3 Report

The topic seems interesting, and the manuscript is well written. However my suggestion for the improvement of quality of this manuscript are given as below.

1)In Introduction section, draw a diagram of mobile communication system and explain it with the help of example.

2) The motivation of this research is missing. Please write it in the introduction section.

3) Please add the future work of your study in conclusion section. 

4) In results section i suggested you to add few diagram by comparing your work with the state of the art algorithms.

Round 2

Reviewer 1 Report

Paper Summary:

The paper studies an effectiveness evaluation method of mobile communication network design schemes and puts forward a design scheme to evaluate and optimize the network plans. It also discusses the details of the improved method of effectiveness evaluation based on factor analysis. Furthermore, the study proposes a decision preference setting method based on cluster analysis to adapt to the preference requirements of different task scenarios. Overall, the authors have tried to present an improved preference selection algorithm, discuss the summary of the evaluation process, and conclude their study. In this regard, the reviewer found the work vital and sound to Generally, the manuscript includes most of the basic sections required for scientific research papers and well-structured that the contents are written in a clear way that is easy to understand the core points. However, to ensure the quality of the paper the following suggestions and comments are needed to be considered and addressed to improve the research further.

Hence, the authors of this paper have tried to incorporate the comments and suggestions provided during our previous review. Considering the improvements made to the paper, the reviewer recommends the following comments as miner issues.

Minor issues:

  1. It can be seen from the body of contents of the abstract section, still the authors have not included the result of an improved preference selection algorithm and the summary of the overall effectiveness of the evaluation process. The reviewer suggests the authors to briefly show the performance analysis result of the solutions and algorithms in the effectiveness evaluation of multiple schemes.
  2. Authors are suggested to insert a separate table for list of notations and common abbreviations that do not need to be defined in expanded text for the sake of clarity to the reader.
  3. There are still few formatting and grammatical errors in some sections of the paper. Hence, authors should recheck, modify and proofread the entire paper before the submission of the paper for publication. The contents of tables, references as well as the conclusion section of this paper should also be updated.
  4. The image used on Figure 7 of the paper is too blurry and still needs better quality (i.e with good DPI quality, appropriate image format like *.png and others.

From the previous review of the paper, the authors have modified and updated the paper as per the suggestions and comments forwarded by the reviewer. If the authors take actions upon the forwarded minor comments above, the reviewer will be satisfied with the quality of the paper and not refuse the submission of the paper for publication.

Reviewer 2 Report

The paper has been revised greatly based on the comments, but the following minor issues should be considered.

1.The following references are related with the topic of this paper, we suggest the authors to discuss them in the revision. Fuzzy Grey Choquet Integral forEvaluation of Multicriteria Decision Making Problems With Interactive and QualitativeIndices,” IEEE Transactions on Systems, Man, and Cybernetics: Systems, doi:10.1109/TSMC.2019.2906635;Green decoration materials selection under interior environment characteristics: a greycorrelationbased hybrid MCDM method,” Renewable & Sustainable Energy Reviews, 2018,81(1): 682-692; "Physical Safety and Cyber Security Analysis of Multi-Agent Systems: A Survey of Recent Advances," in IEEE/CAA Journal of Automatica Sinica, vol. 8, no. 2, pp.319-333, February 2021; "Performance Evaluation of Public Bus Transportation by Using DEA Models and Shannon's Entropy: An Example From a Company in a Large City of China," in IEEE/CAA Journal of Automatica Sinica, vol. 8, no. 4, pp. 779-795, April 2021;and "Safety Evaluation of High Speed Railway LTE-R Communication System Based on AHP and Fuzzy Comprehensive Evaluation," 2019 International Conference on IntelligentTransportation, Big Data & Smart City (ICITBS), 2019, pp. 211-214.

2.The motivation to adopt the present method but not others needs to be presented in detail.

Reviewer 3 Report

The authors has incorporated all necessary changes to the manuscript.

Author Response

Thank you for your review of the manuscript.

This manuscript is a resubmission of an earlier submission. The following is a list of the peer review reports and author responses from that submission.

Round 1

Reviewer 1 Report

Overall, this manuscript is poorly written, missing meaningful discussions and justifications, even a conclusion section. It is also hard to find this work's scientific novelty since most methods used in this work are typical statistical analysis. Notably, the current version of the manuscript looks like a report, not a journal paper.

  • Abstract. The authors need to highlight application domain or problem definitions instead of listing the statistical methods. The authors mainly applied the techniques to a communication system. What can be the authors' contribution?
  • Introduction: The authors need to provide more details about the challenges of effectiveness evaluations for mobile communication systems, the current research work, and the authors' contributions. Simply applying methods to specific datasets cannot be a scientific novelty.
  • Section 2: The methods describe a typical factor analysis and principal component analysis, which can be easily found in textbooks.
  • Section 3: The authors need to explain more about the data sets. For example, the authors may explain how these datasets can be collected and analyzed, what can be application benefit from the datasets.
  • Overall, the table with a lot of numbers is not very interesting.
  • The authors need to write a conclusion section that summarizes the authors' work, contributions, and future work.

Reviewer 2 Report

Dear authors,

I do not detect significant deficiencies of the manuscript in terms of presented formulations and results.

However, before releasing the manuscript, it must be improved as following:

 - check the template of the manuscript and improve it, e. g.  Table 2 overcomes to the next page, it has to also have a caption, lines 233, 234 - it is formulations, which has to be numbered and written in  separate line and similar...

 - in the scientific manuscript, which is supposed to be publish in such a high rated journal, the Sections "Discussion" and "Conclusion" are mandatory.

The Conclusion has to include the summarization of the presented research without any figures

Reviewer 3 Report

The topic can be interesting. However, there are several weakness in this work, such as:

a)First and foremost, I cannot clearly see any new contribution of the current proposed paper.

b) Even though the vocabulary is technical, the ideas, concepts and methodology are presented through very confusing sentences. The paper is poorly organized. There are many grammatical issues/problems and also typos throughout the text as well as in some equations. All these problems in the text make it difficult to follow the stream of thoughts presented by the authors diminishing the quality of the paper and the contribution of the study becomes debatable.

c) There is a non-adequate list of references. More specifically, important references with content relevant to that of the manuscript under consideration are not cited.

d) Statistical methods/software used must be clearly mentioned, justified and discussed.

e) The quality and/or size of the graphs (Figs. 1 -6) presented should be improved.

f) It is not clear how and why the proposed manuscript is different when compared with those involving similar proposed found in the literature. More detailed description and comparison with previous works is needed.

g) The discussions about the results obtained are poor and the conclusion presented are not enough supported by the methods used.

These facts, at my point of view, greatly weakens the manuscript proposed.

Therefore, I suggest to the authors that they rewrite (almost entirely) the manuscript, in order to submit it again.

Reviewer 4 Report

  1. The authors use the concept of effectiveness by some sort of weighting/scoring for communications systems based on the “factor analysis”. Different irrelevant metrics were considered in a set of equations to extract the correlation and then to assign a score to them. In my opinion, the paper is not technically sound for mobile communications systems. You can not look at those parameters like scalar data. I appreciate data-driven approaches for mobile communications, but this paper had significant mistakes. On the other hand, defining a score based on eigenvalue decomposition of a matrix is not novel. These are very basic equations in signal processing and linear algebra. Table 1 is meaningless in communications if the authors plan to have them in one systems of equations as mentioned in the manuscript (X1, X2, …). They are totally different concepts and have different units, trade-offs, and their correlation (in the way that was discussed in the paper) is meaningless. I would not recommend this paper for publication.
  2. The manuscript has numerous grammatical errors. I cannot list these countless errors here, but I would strongly recommend the authors to re-format the paper.
  3. Define \epsilon in equation (1).
  4. Double check equation (2) as the a*\epsilon doesn’t look correct, if the authors are trying to write a matrix format of the equations in (1), then it is definitely wrong.
  5. I don’t find anything novel in equations (4)-(15). Standard normalization is what is being discussed in many textbooks for years as defined in (4).
  6. From equations (1) and (2), index m was changed to n in the rest of equations.
  7. In equations (6), (7), and (9), clarify why “n-1” was mentioned in the denominator instead of “n”. It should be "n" in my opinion.
  8. Table (1): incorporating all those concepts in one set of equations is meaningless in communications. Datasets are valuable but they should be meaningful.
  9. The data in Table (2) is not understandable. 

Round 2

Reviewer 1 Report

The authors improve the quality of the manuscript after the first round of the review. This reviewer provides some minor suggestions, which are the following.

  • In the revised abstract, the authors argue that the application domain of this paper is difficult without providing solid justification. How combat tasks and application environments make the effectiveness evaluation difficult? Why it is difficult to evaluate the effectiveness of network design before application? What does mean by applications?
  • Overall, the authors need a clear distinction between matrixes and scalar variables. Use boldface for matrixes and vectors, not italic. The authors also need to provide the dimensions of the matrixes
  • Equation 12 does not have a proper bound for the index i.
  • When enumerating items, be consistent with a format.

Reviewer 3 Report

Not all changes in the manuscript are marked in any reasonable way, which complicates the evaluation of the revised version. In 'Abstract', in 'Introduction' section (lines 27-29) and (lines 59-83) and other parts of the manuscript including in 'References', there are several changes not marked adequately.
The authors leave the referees to compare original and revised manuscript line-by-line.
Nonetheless, after a new reading, I recognize the authors' effort to provide the responses and corrections requested. Indeed, the paper improved and its contribution is clearer now. 

Reviewer 4 Report

Please note that before getting into any optimization or "decision preference" techniques which we have plenty of them in communications, 1- we need to justify the novelty of the proposed method in terms of framework. I am still not convinced the idea is novel.  2-we need to provide  meaningful indicators. As I said in the previous round, Table 2 doesn't make sense to me as these parameters can not be compared in this way you put them along side each other. I can not analyze the numbers I see in the table and there is no reference for the dataset to refer. 
